# Integrating Spirituality in Group Psychotherapy with First Responders: Addressing Trauma and Substance Misuse

**Caroline Cecil Kaufman** *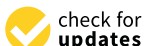, **David Hillel Rosmarin and Hilary Connery**

McLean Hospital, Harvard Medical School, Belmont, MA 02478, USA
* Correspondence: cckaufman@mclean.harvard.edu

**Abstract:** First responders (e.g., fire fighters, law enforcement, paramedics, corrections officers) are at disproportionately high risk of experiencing posttraumatic stress and engaging in substance misuse. Spirituality is a potential source of resilience and recovery in the context of trauma and substance misuse; however, evidence-based clinical protocols integrating spirituality into group psychotherapy with first responders are rare. This article describes the adaptation of an existing and previously examined spiritually integrated group psychotherapy clinical protocol to address substance misuse among first responders with posttraumatic stress. This brief (90-min) and stand-alone group psychotherapy intervention includes (a) psychoeducation about the co-occurrence of trauma syndromes and substance misuse among first responders, (b) discussion of the relevance of spirituality to both posttraumatic stress and substance misuse, and (c) the integration of spiritual beliefs and behaviors to cope with symptoms related to trauma exposures and substance misuse. We discuss relevant clinical theory behind the development of this intervention as well as the potential clinical application of this protocol.

**Keywords:** alcohol; substance use; spirituality; psychotherapy; first responders; trauma; post-traumatic stress

## 1. Introduction

There are unique and unmet mental health needs among first responders (e.g., fire fighters, law enforcement, paramedics, corrections officers) as evidenced by high rates and co-occurrence of alcohol use disorder (AUD) and posttraumatic stress disorder (PTSD). Indeed, rates of alcohol misuse range from 34% to 58% and (Jones 2017) and rates of PTSD range from 15% to 26% among first responders (Jones et al. 2018; Kessler et al. 2005). The co-occurrence of PTSD and AUD is associated with poorer treatment outcomes and poorer treatment completion compared with treatment of either disorder alone, and specialized psychotherapies to effectively treat this combination are not widely accessible (Roberts et al. 2015). Among firefighters, PTSD severity is positively associated with alcohol use coping motives (Lebeaut et al. 2020), which suggests that first responder occupational trauma elevates risk for alcohol misuse and the development of AUD.

First responders, as psychiatric patients, are unique. Firstly, first responders experience recurrent occupational exposures to potentially traumatizing events (e.g., witnessing violence, being assaulted, experiencing line-of-duty injuries, becoming trapped in dangerous circumstances, experiencing line-of-duty deaths of colleagues) (Cheng et al. 2018). Secondly, all first responders are highly exposed to suicide and suicidal behaviors and have greatly elevated risk for suicide themselves (Cheng et al. 2018). Thirdly, first responders may experience a cumulative and "spiraling" of trauma symptoms that aggregates over the course of a career as a first responder (Papazoglou 2013). Fourthly, there is a unique culture among first responders regarding mental health and mental health treatment that values self-reliance and personal strength (Bowers et al. 2022; Jones et al. 2020; Lanza et al. 2018). Finally, first responders identify specific barriers to mental health treatment, including

public stigma, self-stigma, and label avoidance (Bowers et al. 2022; Haugen et al. 2017) as well as mistrust of non-first responders (Kronenberg et al. 2008) and mental health professionals (Repper and Carter 2011). As such, first-responder specific group psychotherapy, as opposed to individual psychotherapy, is often preferred by first responder patients in that it provides opportunities for peer-to-peer support which may reduce perceptions that experiencing psychological distress and seeking mental health/substance use treatment is a sign of weakness (Horan et al. 2021). Despite the unique culture and risks among first responders, fewer than 0.2% of controlled trials of PTSD treatment focus on first responders (Haugen et al. 2012). Consequently, there is a need for the development of targeted treatments specifically designed for first responders as well as research on the efficacy of such treatments (Lanza et al. 2018; Jones et al. 2020).

Spirituality is an important aspect of identity and experience for first responders (Arble et al. 2018; Moran 2017) and may also be of special importance among individuals with AUD and/or PTSD (e.g., McCormack and Riley 2016; Papazoglou et al. 2020). Spirituality is broadly defined as one's connection with humanity, the sacred, or the soul and one's sense of purpose (Harris et al. 2018). In accordance with this definition, spirituality may be connected or unconnected with one's religion and religious practices. In addition, spirituality may be considered an ineffable aspect of culture and identity for some individuals (Tisdell and Tolliver 2003). Spiritual effort (i.e., effortful engagement in spiritual practices), engagement in spiritual practices, and spiritual growth are associated with reduced risk of alcohol misuse among police officers (Chopko et al. 2016), and are also associated with posttraumatic growth and better coping in first responders experiencing occupational trauma (Lentz et al. 2021; Smith 2009). A spiritually integrated intervention (i.e., protocol with spiritual elements) for first responders demonstrated improvements in mental health (i.e., reductions in anxiety and depressive symptoms), but PTSD and AUD symptoms were not specifically targeted or assessed in this study (Knobloch and Owens 2021). Additional research is needed to explore applicability of spirituality to interventions with first responders as well as perceived benefit and efficacy of such interventions.

Spirituality may be a particular salient aspect of identity and experience among individuals experiencing PTSD and/or AUD. For example, daily experiences of spirituality are protective against poor mental health (i.e., depression and anxiety) and interpersonal outcomes in persons with PTSD (Blakey 2016; Kaufman et al. 2020, 2021). Spirituality is also a source of resilience and coping among individuals with either AUD (Dermatis and Galanter 2016; McInerney and Cross 2021) or PTSD (Currier et al. 2015). Theory in this area of research suggests that individuals may utilize spiritual beliefs/practices as a source of hope, emotion regulation, and meaning-making in the context of trauma or AUD (Starnino 2016) because it provides individuals with "a structure for understanding the world and events that occur" and may provide "a mechanism to transcend events of this life" (Farley 2007, p. 4). Supporting this theory, veterans with higher levels of spirituality benefited more from residential treatment than did veteran peers with lower levels of spirituality (Currier et al. 2019).

A few notable spiritually integrated interventions have been studied in first responders and have demonstrated improvements in important aspects of mental health (e.g., anxiety and depressive symptoms) and physical health (e.g., fatigue, sleep) (e.g., Knobloch and Owens 2021; Thompson and Drew 2020). For example, the REBOOT program provides Christian-based, peer-led group services to first responders and their families (Knobloch and Owens 2021). The program involves spiritually based group- and individual-based reflection, journaling, and discussion activities to address distress and symptoms related to serving as a first responder (Knobloch and Owens 2021). However, this intervention was "non-clinical" and Christian centric in nature, and researchers did not assess PTSD or AUD symptoms. Additionally, Thompson and Drew (2020) examined the impact of a 21-day resilience training among first responders that included core elements of spiritual practice, including gratitude, meaning making, and overall life purpose. The training included daily activities that were completed individually by first responders, including journaling

and reflection. Results indicated improvements in gratitude and perceived emotional stability; however, up to 31% of the sample denied changes in outcomes (Thompson and Drew 2020) and outcomes specific to AUD or PTSD were not analyzed. As such, there is a need the development of protocols that may result in greater rates of change among first responders. Given the prevalence and consequences of co-occurring PTSD and AUD among first responders, and the paucity of spiritual mental health research in this population, we aimed to develop a spiritually integrated treatment intervention to address this treatment gap.

A spiritually integrated psychotherapy protocol has been developed for psychiatric patients in inpatient, residential and acute settings (Rosmarin et al. 2011b, 2019, 2021), which was well-tolerated and perceived as beneficial across psychiatric patients, including those with diagnosis of AUD or/and PTSD (Rosmarin et al. 2019, 2021). This protocol, called Spiritual Psychotherapy for Inpatient, Residential, and Intensive Treatment (SPIRIT) incorporates a spiritual framework into cognitive behavioral therapy (CBT) group protocol (Rosmarin et al. 2011b, 2019). To address the unmet need for spiritually integrated psychotherapy targeted for first responders with co-occurring AUD and PTSD, we adapted SPIRIT to focus on AUD and PTSD symptoms and delivered a pilot series to first responders receiving acute mental health/substance use care, calling this adaptation SPIRIT for First Responders (SPIRIT-FR).

## 2. Treatment Setting

SPIRIT was previously developed and disseminated within McLean Hospital, a private psychiatric hospital located in Belmont, MA and an affiliate of Harvard Medical School. McLean Hospital has a unique treatment program for first responders, called the LEADER (Law Enforcement, Active Duty, Emergency Responder) Program. The LEADER Program provides services to first responders with any mental health/substance use diagnoses and accommodates levels of care to serve illness severity (including inpatient, acute residential, partial hospital, and outpatient). Culturally sensitive care is prioritized across treatment within the LEADER Program (Lewis-Schroeder et al. 2018). Specifically, clinicians remain curious and aware of cultural beliefs, practices, and realities (e.g., shift work, barriers to access to care, burnout, stigma) associated with serving as a first responder.

## 3. Present Centered Approach

The SPIRIT-FR protocol is implemented with a present centered approach (Schnurr et al. 2003). The present centered approach is derived from present centered therapy (PCT), an evidence-based treatment for PTSD among military personnel (Frost et al. 2014; Schnurr et al. 2003, 2007). PCT focuses on grounding patients in the present moment, altering maladaptive thoughts and behaviors, teaching problem-solving strategies related to the impact and intersection of trauma and substance use on the patient's current functioning, and psychoeducation (Najavits 2002; Schnurr et al. 2003, 2007). The treatment paradigm of PCT differs from trauma-focused approaches that include elements of exposure and re-experiencing (Schnurr et al. 2003, 2007). Meta-analytic results indicate PCT is as good or better than trauma-focused treatments for reduction of PTSD symptoms in active-duty military populations (Belsher et al. 2019; Frost et al. 2014). Similarly, PCT for substance use disorders show significant improvements post-treatment and at follow-up among women with PTSD (Hien et al. 2010). In addition, PCT for PTSD and co-occurring PTSD and AUD shows lower dropout rates among adults and veterans when compared with trauma focused therapies (Belsher et al. 2019; Capone et al. 2020; Imel et al. 2013). The SPIRIT-FR protocol utilizes a present centered approach that acknowledges that each patient has experienced trauma and potentially long-standing patterns of alcohol misuse but asks that patients do not share the particulars of experienced traumatic events and/or the origins of their substance use in order to focus on the present and to avoid exposure elements that might be trigger exacerbation of symptoms.

## 4. Treatment Description

The original SPIRIT protocol is based on a CBT framework that includes psychoeducational components as well as specific spiritual skills, activities, and beliefs (Rosmarin et al. 2019, 2021). SPIRIT is delivered in a single group session format and was designed for implementation with acute care mental health/substance use patients. SPIRIT-FR is likewise a single-session, spiritually integrated group psychotherapy protocol for first responders with co-occurring PTSD and AUD, and the content is delivered flexibly so that clinicians can adjust content to the group-specific needs.

SPIRIT-FR is a departure from 12-Step programs because it does not require participants to believe in G-d and/or a Higher Power and also does not necessarily promote abstinence as the goal of treatment (Alcoholics Anonymous 2001). In addition, unlike 12-Step programs, the current protocol is designed to recognize the potential role of post-traumatic stress in alcohol misuse and is also specifically designed for delivery in acute psychiatric settings with first responders.

SPIRIT-FR is divided into four parts over the course of one, 90-minute session (see Appendix A). Part I begins with an ethical disclaimer from the clinician clarifying that the purpose of the group is not aimed at spiritual/religious conversion and that all spiritual/religious beliefs and practices will be equally respected, including atheist beliefs. Consistent with first responders' concerns about stigma (Bowers et al. 2022; Haugen et al. 2017), Part I also includes a brief discussion of the importance of confidentiality and privacy among patients. Finally, Part I identifies the group as being part of McLean Hospital's Spirituality and Mental Health Program, a program to provide patients with spiritually integrated and evidenced-based care. Part I lasts five to seven minutes.

Part II of the group provides patients with psychoeducation regarding the frequent co-occurrence of trauma and alcohol misuse among first responders. This part of the group is introduced using the follow phrase: "We are going to take several minutes to consider and discuss how PTSD symptoms may influence alcohol use and vice versa." Subsequently, the clinician provides Handout 1 (Appendix A) to stimulate discussion regarding the relationship between PTSD symptoms and alcohol use among first responders. Using the handout and informed by the approach/avoidance coping model (Arble and Arnetz 2017), patients are encouraged to consider how alcohol may be ineffectively used to avoid and/or quell PTSD symptoms, which, in turn, reinforces and/or strengthens PTSD symptoms (Arble and Arnetz 2017; Tripp et al. 2020). Clinicians also encourage discussion about how and why this relationship might be particularly common among first responders (e.g., drinking culture, stigma). Potential leading questions for this discussion include: How are your symptoms of PTSD (e.g., hyperarousal, mood, cognitions) connected to your history of or current alcohol use? What kind of relationship do you notice between cravings for alcohol and PTSD symptoms? How does alcohol use impact your symptoms of PTSD (e.g., hyperarousal, mood, cognitions)? Throughout the discussion, the clinician normalizes the bidirectional relationship between PTSD symptoms and alcohol misuse to provide perspective aimed at reducing personalization and feelings of shame, stigma, and guilt (Fay et al. 2006; Jones et al. 2020).

During Part II, the clinician also asks patients to reflect on how their own symptoms of PTSD may influence cravings and use of alcohol (Worksheet 1). The clinician leads the group through the example provided on the worksheet and answers any questions asked by group members about this example. This example is informed by research indicating that PTSD, sleep disturbances, and alcohol use intersect among first responders (Smith et al. 2018). Next, the clinician asks patients to take several minutes to complete the worksheet to help patients identify relationships across their own experience of PTSD symptoms and cravings for and actual use of alcohol to cope with and/or avoid symptoms. Patients are asked to leave the final column (i.e., Alternative way(s) of coping) blank until Part IV of the session. In total, Part II of the session is expected to take 30 min.

During Part III of the session, the clinician asks participants to reflect upon and consider how spiritual beliefs, traditions, and rituals may or may not relate to their symptoms

of PTSD and/or AUD. The clinician provides the following definition of spirituality to patients: thoughts, feelings, and behaviors having to do with a person's search for the sacred or transcendent (Pargament 2007; Rosmarin et al. 2011a). Examples of leading questions include: How is your spirituality (e.g., beliefs, traditions, practices) relevant to your PTSD and/or AUD symptoms? How do you draw upon your spirituality when understanding your symptoms? What messages have you received from your spiritual tradition or community about your symptoms? These questions are adapted from Rosmarin (2018) and Rosmarin et al. (2019). Throughout the discussion, the clinician should validate patients' experiences, including if patients deny a relationship between their spirituality and symptoms. Part III of the session is anticipated to take 25 min.

Finally, Part IV of the session provides patients with alternative ways of coping using both cognitive and behavioral spiritual resources. Just as in the original SPIRIT protocol, this part of the group is introduced with the following phrase: "Irrespective of how spirituality relates to your personal mental health, it can be helpful to draw upon spiritual resources in shaping our thoughts, behaviors, and feelings" (Rosmarin et al. 2019). Subsequently, the clinician provides patients with Handout 2 (see Appendix A) that details nine spiritual/religious coping skills. This list of skills is adapted from the original SPIRIT protocol (Rosmarin et al. 2019) and is informed by prior research examining the intersection between PTSD and alcohol use among first responders and community members. Specifically, awareness and relaxation activities (i.e., mindfulness, prayer, meditation, and yoga) are associated with lower PTSD symptom severity and stress among police officers and psychiatric inpatients (Arble and Arnetz 2020; Martin et al. 2018). Similarly, meaning making (i.e., sacred texts, finding the meaning, and counting your blessings) is associated with changes in cognition, mood, and behavior (Pennebaker 1997). Finally, social support (i.e., seek religious support) and self-compassion are associated with better functioning in the context of PTSD among veterans and first responders (Arble and Arnetz 2020; Kshtriya et al. 2020).

The clinician provides an overview of the spiritual coping resources and provides brief definitions and examples of each. The clinician then facilitates a group discussion by asking questions, including: Which of these activities have you done in the past or do you currently use? How do they impact your alcohol use? How do they impact your urges to drink? Are there any activities on this list that you would be interested trying? At the end of the discussion, the clinician asks patients to return to Worksheet 1 to add an alternative coping skill to the final column. Part IV of the session is anticipated to take 25 min.

The session closes with a brief summary from the clinician and expressed gratitude towards patients for their willingness to participate in the session. The closing of the session is anticipated to take two to five minutes.

## 5. Discussion

Spirituality is an identified source of resilience in both first responders and the general population having PTSD or alcohol/substance use (Chopko et al. 2016; McCormack and Riley 2016; Papazoglou et al. 2020). Intersecting risk factors highlight the need for tailored and targeted treatment among first responders with mental health/substance us disorders. These factors include high rates of co-occurring PTSD and AUD (Jones 2017; Jones et al. 2018; Kessler et al. 2005), unique cultural norms in first responder communities (Bowers et al. 2022; Haugen et al. 2017; Kronenberg et al. 2008; Repper and Carter 2011), and the lack of spiritually integrated psychotherapy to address alcohol misuse among first responders specifically (Ralevski et al. 2014; Simpson et al. 2021). Consequently, we adapted the previously disseminated SPIRIT protocol (Rosmarin et al. 2019, 2021) into this single-session group psychotherapy protocol for first responder patients with co-occurring PTSD and AUD. SPIRIT-FR would be expected to expand clinical treatment options for first responder patients and expand the available options for spiritually integrated psychotherapy.

Spiritually integrated psychotherapy and mental health care for first responders requires clinician training in (1) the intersection of spirituality and mental health care delivery (Rosmarin et al. 2019), and (2) cultural competence in understanding first responder occupational structures and training that shape beliefs about seeking mental health/substance use care (Bowers et al. 2022; Haugen et al. 2017; Jones et al. 2020; Kronenberg et al. 2008; Lanza et al. 2018; Repper and Carter 2011). It may be difficult for clinicians to gain access to this training, as most training programs do not require any coursework or supervised practice in spirituality and mental health (Vieten and Lukoff 2022) and few offer opportunities to train specifically on first responder culture. The SPIRIT protocol and the current adapted SPIRIT-FR attempt to address this gap in training by providing handouts and guidance for clinicians, however, this may not be sufficient to competently implement this protocol (Rosmarin et al. 2021), and therefore we recommend additional training resources included in the Appendix A.

As stated previously, to our knowledge, this is the first single-session clinician-led protocol for spiritually integrated psychotherapy designed to address alcohol misuse and PTSD symptom management among first responders with co-occurring PTSD and AUD. There are certain limitations to the protocol, including the current lack of feasibility, acceptance, and efficacy testing of this session. We plan to examine feasibility and perceived benefit(s) in future research with patients at the LEADER Program at McLean Hospital with the aim of providing a holistic care approach to complex mental health/substance use disorders.

**Author Contributions:** Conceptualization, C.C.K., D.H.R. and H.C.; resources, C.C.K., D.H.R. and H.C.; writing—original draft preparation, C.C.K.; writing—review and editing, H.C. and D.H.R.; supervision, D.H.R.; project administration, D.H.R.; funding acquisition, D.H.R. and H.C. All authors have read and agreed to the published version of the manuscript.

**Funding:** This research received no external funding.

**Institutional Review Board Statement:** Not applicable.

**Informed Consent Statement:** Not applicable.

**Conflicts of Interest:** The authors declare no conflict of interest.

### Appendix A

Clinical Protocol for Integrating Spirituality in Group Psychotherapy with First Responders: Addressing Trauma and Substance Misuse.

| Group Title |
| --- |
| • Integrating Spirituality in Group Psychotherapy with First Responders: Addressing Trauma and Substance Misuse |
| **Length** |
| • 75–90 min session |
| • Stand-alone group session for inpatient/residential settings with first responders with comorbid PTSD and AUD |
| **Objectives of Group** |
| • To help patients understand the frequent co-occurrence of trauma and substance misuse. |
| • To help patients explore/understand relationships between their spirituality and PTSD/AUD. |
| • To help patients identify spiritual beliefs and behaviors they can utilize to cope with trauma symptom management and alcohol use. |
| **Handouts** |
| • Handout 1: Relationship between PTSD and Alcohol Misuse |
| • Handout 2: Spiritual/Religious Coping Skills |
| **Worksheet** |
| • Worksheet 1: Understanding Alcohol Use and Alternative Ways of Coping |

**Outline of Group and Timing of Handouts**

*Part I: Introduction to Group and Confidentiality and Privacy among Patients—3–5 min*

- Context: This group is a part of the McLean Hospital Spirituality Program, a hospital-wide initiative to provide patients with spiritually integrated, evidence-based care.
- Disclaimer (read to patients): The purpose of this group is not to convert you or to give patients or clinicians an opportunity to convert others. Along these lines, people have different faiths and traditions, so please be respectful to everyone's beliefs and practices.
- Confidentiality and Privacy: Everything that is said within the group is private and confidential. Group members should not discuss what is said within the group to others (e.g., patients, friends, colleagues).

*Part II: Overlap of Trauma and Alcohol Use Among First Responders (Using 1 Handout and 1 Worksheet) 30 minutes*

- Introduce the section: We are going to take several minutes to consider and discuss how PTSD symptoms may influence alcohol use and vice versa.
  - **Questions: How are your symptoms of PTDS (e.g., hyperarousal, mood, cognitions) connected to your history of or current alcohol use? What kind of relationship do you notice between cravings for alcohol and PTSD symptoms?**
  - Guide the discussion such that each patient has an opportunity to share how their alcohol use may relate to their PTSD symptoms. Try to find common themes among patients and point those out to the group.
- Provide patients with Handout 1: Relationship between PTSD and Alcohol Misuse
  - Frame for patients (psychoeducation): research suggests that individuals may use alcohol to avoid or reduce their PTSD symptoms.
  - However, using alcohol to reduce or avoid PTSD symptoms (e.g., hyperarousal, trouble sleeping, mood, re-experiencing) may actually cause PTSD symptoms to persist and worsen.
  - Allow patients to discuss this information. Throughout the discussion, highlight when patients identify the bidirectional relationship between PTSD and alcohol misuse.
    - **Question: Why or how might this relationship be especially true for first responders?**
- Provide patients with Worksheet 1: Understanding Alcohol Use and Alternative Ways of Coping
  - Introduce the worksheet: We are going to consider how our own symptoms of PTSD may influence cravings of or use of alcohol.
  - Lead patients through the example provided.
  - Ask patients to complete the worksheet on their own, but to leave the final column (i.e., Alternative Way(s) of Coping) blank. Provide an opportunity for 1–2 patients to share their example.

*Part III: Spirituality's Relationship with PTSD and Alcohol Use—25 min*

- Introduce the section: For many people, spirituality plays a role in how they experience and understand their symptoms of PTSD and alcohol use.
  - Define spirituality: For our group, spirituality is defined as thoughts, feelings, and behaviors having to do with a person's search for the sacred or transcendent.
- **Questions: How is your spirituality (e.g., beliefs, traditions, practices) relevant to your PTSD and/or AUD symptoms? How do you draw upon your spirituality when understanding your symptoms? What messages have you received from your spiritual tradition or community about your symptoms?**
  - Allow patients to share their experiences and encourage patients to find commonalities across their experiences
  - Validate all patients' experience and even those that deny any relationship between their spirituality and symptoms.

*Part IV: Spiritual/Religious Coping Skills (Handout 2: Spiritual/Religious Coping Skills) 25 min*

- Introduce the section: Irrespective of how spirituality relates to your personal mental health, it can be helpful to draw upon spiritual resources in shaping our thoughts, behaviors, and feelings.
- Instruct patients: To these ends, here is a handout with a list of common spiritual/religious coping skills that can be used to cope with symptoms of PTSD and cravings for alcohol use. Some of these examples may not be appropriate or of interest to you.
- Provide patients with Handout 2: Spiritual/Religious Coping Skills.
- **Questions: Which of these activities have you done in the past or currently use? How do they impact your alcohol use? How do they impact your alcohol urges and use? Are there any activities on this list that you would be interested trying?**
  - Using these questions, facilitate a discussion among patients.
- Ask patients to return to Worksheet 1: Understanding Alcohol Use and Alternative Ways of Coping.
  - Instruct patients to add a spiritual/religious coping skill that they would be willing to utilize to final column for the example they generated earlier in the discussion.
- Ask 1–2 patients to share the spiritual/religious coping skill they applied in the final column of the worksheet.
  - Encourage patients to identify at least one skill they would be willing to use in the future.

| | |
|---|---|
| *Conclusion—2–5 min* | |
| • Gratitude: Thank patients for their willingness to participate in the session and share their experiences. | |
| • McLean Hospital is proud to provide you with this group and other opportunities for spiritually integrated care. | |
| • If anyone would like to speak one-on-one about spirituality/religion, our hospital inter-faith chaplain is available. Please let your treatment team know so that they can request a consultation for you. | |
| • We encourage you to speak to your treatment providers about spiritual/religious issues as they pertain to your symptoms and treatment, both while you're at the hospital and as part of your discharge plan. | |

**Worksheet 1: Understanding Alcohol Use and Alternative Ways of Coping.**

| Trigger<br>*What PTSD symptom and/or situation set me up to have an urge or use?* | Thoughts and Feelings<br>*What was I thinking? What was I feeling?* | Behavior/Urge<br>*What did I do or want to do?* | Outcome(s)<br>*What positive and/or negative things happened?* | Alternative Way(s) of Coping |
|---|---|---|---|---|
| **Example:** *I was lying in bed and couldn't fall asleep. I felt worried about having nightmares while asleep and couldn't calm down enough to sleep.* | **Example:** *I'll never be able to fall asleep and if I do, I'll have a nightmare and wake up feeling even worse.* | **Example:** *Drink a couple of beers until I'm sleepy enough to fall asleep.* | **Example:** *I was able to fall asleep but had a nightmare anyway. In the morning, my spouse found the empty beer cans and threatened to leave again.* | **Example:** *Read passages from a sacred text (e.g., Bible) until I am ready to fall asleep.* |

Handout 2: Spiritual/Religious Coping Skills

Many people draw upon spiritual/religious beliefs, attitudes, or practices to reduce emotional distress. Below are some examples of spiritual/religious activities that you may wish to use to cope with urges to use alcohol or other substances.

| |
|---|
| *Mindfulness*<br>Engage in a mindfulness practice via a recorded activity or by paying attention to the five senses in your body.<br><br>*Prayer*<br>Prayer involves speaking from the heart to one's Higher Power. Prayer can be formal and structured, or spontaneous.<br>Here are four types of prayer:<br>(1) Thanks—e.g., "Thank you for the sandwich I had for lunch today"<br>(2) Praise—"It's amazing how many types of apples there are"<br>(3) Conversation—"I feel really angry right now that I got a speeding ticket!"<br>(4) Request—"Please help me to get to my appointment on time"<br><br>*Count your Blessings*<br>Think about three things you are grateful for each day.<br><br>*Yoga*<br>Spend a few minutes practicing yoga.<br><br>*Finding the Meaning*<br>Focus on something that is meaningful and important to you, despite your suffering.<br><br>*Meditate on a Coping Statement*<br>Pick an inspiring quotation that is personally meaningful and write it on an index card, then repeat it over to yourself throughout the day.<br><br>*Sacred Texts*<br>Read or listen to passages from the Bible, Torah, Quran or other sacred texts.<br><br>*Seek Religious Support*<br>Speak to your clergy, family, or friends about spirituality/religion.<br><br>*Self-Compassion*<br>Practice self-compassion by giving yourself permission to take however much time you need to cope with symptoms and urges. |

**Additional Training Resources**
**1. Publications:**

Pargament, Kenneth I. 2007. Spiritually integrated psychotherapy: Understanding and addressing the sacred. Guilford press. (Pargament 2007)

Pargament, Kenneth I., and Julie J. Exline. 2021. Working with spiritual struggles in psychotherapy: From research to practice. Guilford Publications. (Pargament and Exline 2021)

Park, Crystal L., Joseph M. Currier, J. Irene Harris and Jeanne M. Slattery. 2017. Trauma, meaning, and spirituality: Translating research into clinical practice. American Psychological Association. (Park et al. 2017)

Rosmarin, David H. 2018. Spirituality, religion, and cognitive-behavioral therapy: A guide for clinicians. Guilford Publications. (Rosmarin 2018)

Rosmarin, David H., Randy P. Auerbach, Joseph S. Bigda-Peyton, Thröstur Björgvinsson, and Philip G. Levendusky. (2011). Integrating spirituality into cognitive behavioral therapy in an acute psychiatric setting: A pilot study. Journal of Cognitive Psychotherapy, 25(4), 287. (Rosmarin et al. 2011b)

**2. Institutions:**

The Albert & Jessie Danielsen Institute at Boston University.

Center for Spirituality, Theology, and Health at Duke University.

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
