# Peer review of "Integrating Spirituality in Group Psychotherapy with First Responders: Addressing Trauma and Substance Misuse"

_religions, doi:10.3390/rel13121132_

Round 1
Reviewer 1 Report
Page 2, first paragraph – the reference for RCTS focused on first responders is old (Haughen et al., 2012). Do you have any newer references?
Inconsistencies in APA style (Tisdel & Tolliver 2003) has a comma where there shouldn’t be one, some parenthetical citations are not in alphabetical order
Page 3 – the argument that 31% of the sample had no changes in outcomes… I am not sure how this promotes your argument, you may want to consider removing or expanding on your point here.
Can you provide any context about how the therapist should be trained? Should the therapist have some competency/background knowledge in specific PRS concepts, like Pargaments spiritual coping or Exline’s spiritual struggles?
You mention cultural competence of working with first responder culture. What would this look like - what recommendations would you offer for therapists thinking of using this protocol? How does this intersect with other cultures, including spirituality?
Author Response
Page 2, first paragraph – the reference for RCTS focused on first responders is old (Haughen et al., 2012). Do you have any newer references?
We are not aware of another study that reports the number of RCTs including first responders. However, the paper we referenced (i.e., Haughen et al., 2012) has been widely cited since 2012 and continues to be widely cited in the first responder research literature (e.g., Alden et al., 2020; Alshahrani et al., 2022).
Inconsistencies in APA style (Tisdel & Tolliver 2003) has a comma where there shouldn’t be one, some parenthetical citations are not in alphabetical order.
Thank you for this comment. We have made changes to the citations throughout the manuscript to be consistent with APA style (e.g., pgs. 1-5).
Page 3 – the argument that 31% of the sample had no changes in outcomes… I am not sure how this promotes your argument, you may want to consider removing or expanding on your point here.
Thank you for an opportunity to clarify our argument. We included this percentage to highlight that almost one-third of first responders in the intervention cited did not necessarily benefit from the intervention (i.e., lack of changes in outcomes). As such, there is a need for development of interventions in this area with the hope that a larger percentage of first responders benefit from the intervention. To address this point, we have added text to pg. 3 of the manuscript (e.g., “As such, there is a need the development of protocols that may result in greater rates of change among first responders”).
Can you provide any context about how the therapist should be trained? Should the therapist have some competency/background knowledge in specific PRS concepts, like Pargaments spiritual coping or Exline’s spiritual struggles?
As stated on pgs. 4-5 of the manuscript, it may be challenging for clinicians to gain access to training in spirituality and mental health (e.g., “It may be difficult for clinicians to gain access to this training, as most training programs do not require any coursework or supervised practice in spirituality and mental health few offer opportunities to train specifically on first responder culture”). We also highlight the potential need for additional training in this area (e.g., “this may not be sufficient to competently implement this protocol (Rosmarin et al., 2021), and therefore we recommend additional training resources included in the appendix”) on pg. 6 of the manuscript. We also provide additional training resources in the Appendix, including one co-written by Pargament and Exline and one written by Pargament, independently. Clinicians do not require specific knowledge in Kenneth Pargament’s spiritual coping or Julie Exline’s spiritual struggles—however, familiarity with these works could be beneficial.
You mention cultural competence of working with first responder culture. What would this look like - what recommendations would you offer for therapists thinking of using this protocol? How does this intersect with other cultures, including spirituality?
Thank you for this comment. As stated on pg. 3 of the manuscript, relevant “cultural” pieces or realities among first responders may include shift work, stigma towards mental health treatment as well as potential barriers to care (pg. 5). To highlight this point, we added language to pg. 3 of the manuscript to expand upon aspects of the unique cultural aspects among first responders (e.g., barriers to access to care, burnout, stigma). On pg. 5 of the manuscript, we described the need for clinicians to have access to training that facilitates “cultural competence in understanding first responder occupational structures.” To address your point, we added an additional training resource to the Appendix in cultural considerations among first responders (i.e., Kronenberg et al., 2008)
Reviewer 2 Report
Summary and first impression
This paper first defines an unmet need for spiritually integrated psychotherapy targeted for first responders with co occurring alcohol use disorders or post traumatic stress disorders. Then it describes both the content and theoretical background and choices made to adapt an existing spiritually integrated psychotherapy protocol for this specific population.
The strength of this paper is it clear line of reasoning, very well documented, with nice and instructive enumerations like in the second paragraph (p1), and its stepwise and clinically responsible development of a practical psychotherapeutic intervention.
This paper focusses on the application of the SPIRIT-FR in first responders. Some minor points remain unclear with regard to relation of the content to already available interventions in AUD/PTSD and the need of a clinician-led model. It is a practical, thorough and well written paper. After minor revision on the points specified below, I advise acceptance.
Specification
This paper focusses on the application in first responders, and shortly describes other interventions, mostly peer-led, for this group. It is not clear from this article how this protocol with regard to its contents relates to existing AUD / PTSD psychotherapeutic group interventions. The 12 Step Minnesota Model, for example, integrates spirituality and meaning making also. In addition, it is not so clear what the additional value is to make this a clinician-led intervention – especially given the reported mistrust of non-first responders and mental health professionals (p2).
Additional details
- Page 1, main text, second paragraph: ‘Secondly, all first responders are highly exposed to suicide..’ Add ‘and suicidal behaviors’?
- Some spaces are double, where one would expect single spaces. E.g. in third line page three.
- This article is well referenced, sometimes perhaps somewhat abundantly. E.g. the reference to Verplaetse 2018 in the second paragraph of the discussion (p5) seems not to support the content of this sentence. Verplaetse does not mention spirituality or meaning making anywhere in the given paper.
Author Response
This paper focusses on the application in first responders, and shortly describes other interventions, mostly peer-led, for this group. It is not clear from this article how this protocol with regard to its contents relates to existing AUD / PTSD psychotherapeutic group interventions. The 12 Step Minnesota Model, for example, integrates spirituality and meaning making also.
Thank you for these comments. The 12-step program and other programs such as the 12 Step Minnesota Model have been helpful and powerful for many individuals, including first responders. However, these programs do not clearly focus on the intersection of posttraumatic stress and alcohol misuse and are not specifically designed for delivery in an acute psychiatric setting.
Because first responders particularly struggle with this intersection, there is a need for treatment that recognizes both traumatic symptoms and alcohol misuse among this unique population. As such, our intervention expands upon prior work by incorporating spirituality, meaning making, trauma, and alcohol misuse in an acute psychiatric setting (pg. 3). To address this point, we have added language to pg. 4 of the manuscript (e.g., “SPIRIT-FR is a departure from 12-Step programs because it does not require participants to believe in G-d and/or a Higher Power and does not necessarily promote abstinence as the goal of treatment (Alcoholics Anonymous, 2001). In addition, unlike 12-Step programs, the current protocol is designed to recognize the potential role of posttraumatic stress in alcohol misuse and is also specifically designed for delivery in acute psychiatric settings”).
Although peer-led interventions referenced in our paper, these interventions were either Christian-specific or did not result in outcome change for a significant portion of participants (pgs. 2-3). As such, there is a need for interventions that are not specific to the Christian religion and that may result in outcome change for a larger number of participants.
In addition, it is not so clear what the additional value is to make this a clinician-led intervention – especially given the reported mistrust of non-first responders and mental health professionals (p2).
Thank you for this comment. Because this protocol is designed specifically for first responder psychiatric patients receiving acute care, it is appropriate that this protocol be led by a trained mental health clinician.
Page 1, main text, second paragraph: ‘Secondly, all first responders are highly exposed to suicide..’ Add ‘and suicidal behaviors’?
Thank you for this comment “and suicidal behaviors” has been added to pg. 1 of the manuscript.
Some spaces are double, where one would expect single spaces. E.g. in third line page three.
Thank you for noticing this problem. We will rely on the editing team, whose template this is, to fix this issue.
This article is well referenced, sometimes perhaps somewhat abundantly. E.g. the reference to Verplaetse 2018 in the second paragraph of the discussion (p5) seems not to support the content of this sentence. Verplaetse does not mention spirituality or meaning making anywhere in the given paper.
We appreciate note of this. We have removed this reference from pg. 5 of the manuscript.